# Feasibility of up-tower repair concepts for pultruded carbon spar cap planks in wind turbine blades

Malo Rosemeier[1], Matthias Saathoff[2], Johannes Kolsch[3], Pablo Buriticá[4], Enno Petersen[1], and Christopher Beer[2]

[1]Department of Rotor Blades, Fraunhofer IWES, Fraunhofer Institute for Wind Energy Systems, Am Seedeich 45, 27572 Bremerhaven, Germany
[2]apera engineers GmbH, Wendenstrasse 130, 20537 Hamburg, Germany
[3]Renertech Betriebsführungs GmbH & Co. KG, Graf-Zeppelin-Strasse 68, 33181 Bad Wünnenberg, Germany
[4]DNV Renewables Certification GmbH, Brooktorkai 18, 20457 Hamburg, Germany

**Correspondence:** Malo Rosemeier (malo.rosemeier@iwes.fraunhofer.de)

**Abstract.** While modern wind turbine blades utilize pultruded carbon fiber-reinforced polymer (CFRP) planks for structural integrity in spar caps, these materials can sustain damage from operational stresses, leading to potential failures if unaddressed. Traditional down-tower repairs result in significant costs related to dismantling and transportation, especially for offshore installations, emphasizing the need for efficient up-tower repair methods. The research utilizes a finite element model of an 81.6 m rotor blade designed for a 7 MW offshore turbine, subjected to aeroelastic simulations to evaluate load conditions during maintenance. The analysis focuses on a step-wise increased repair zone, assessing susceptibility to buckling, cyclic strains, and permissible wind speeds. Results indicate that while substantial repairs can endanger structural stability, turbulence-induced strain amplitudes are manageable. Recommendations include installing temporary pretensioning and buckling support structures to enhance safety during repairs. Various innovative support concepts are proposed for installation from both inside and outside the blade, aimed at improving structural integrity during up-tower repairs.

## 1 Introduction

Modern wind turbine blades utilize carbon fiber-reinforced polymer (CFRP) planks in their load-carrying structures, i.e., the spar caps. These CFRP planks are pre-manufactured in a pultrusion process and stacked into the blade structure. The pultrusion process ensures high quality fiber alignment and fiber volume fraction, and low numbers of manufacturing defects. It can further ensure that all these properties are produced with high reproducibility, leading to outstanding mechanical properties. CFRP planks can be damaged during operation due to stress concentration resulting from insufficient bonding between planks, voids, defects, dielectric breakdowns (Harrell, 2020), or flashovers caused by a lightning strike (Girolamo, 2020). If such damage is not repaired, it can lead to blade loss (Fig. 1).

When CFRP planks near the inner blade surface are damaged in the inboard blade region (Fig. 2), the repair must be carried out from the inside, whereby the damaged CFRP planks must be ground off and a window cut out of the shear web (Fig. 3). Currently, a repair of a damaged area on pultruded CFRP spar caps is carried out by removing the affected rotor blade from

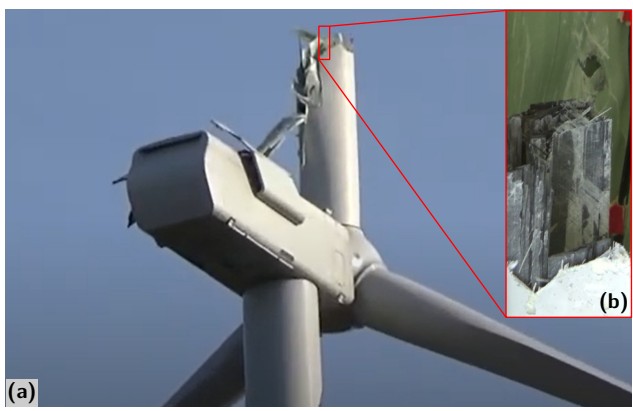

**Figure 1.** Blade loss incident after 2 years of operation (Pierrot, 2022) showing typical damage resulting from a lightning strike event; Ylikoski (2020).

the wind turbine, i.e., performing a down-tower repair, whereby suitable supports are used to relieve the load on the damaged structure during repair. This can result in significant costs for dismantling/assembly, transportation, and downtime, depending further on whether the wind turbine is operated on land or at sea, e.g., on a floating or fixed platform. Therefore, it is desirable
to provide repair and preparation methods, as well as devices and systems suitable for use in such methods, that can reduce such costs. Consequently, an up-tower repair is generally preferred over a down-tower repair on site or in the workshop. In an up-tower repair situation, the temporarily weakened structure needs to withstand extreme wind situations, particularly offshore. To the authors' knowledge, no existing up-tower repair concepts for CFRP planks have been identified in the current literature.

To this end, this research investigates the feasibility of an up-tower repair concept by means of (i) simulations and (ii)
practical considerations. Therefore, a finite element (FE) model of an 81.6 m long commercial rotor blade, which was designed for a 7 MW offshore turbine, was subjected to a maintenance load situation. The blade loads were determined by aeroelastic simulations. The weakened blade structure, i.e., a step-wise increased repair zone, was investigated in respect of buckling, cyclic strains, and the permissible wind speed for conducting the repair.

This article is structured as follows: Sec. 2 presents the suggested up-tower repair concept and describes how the loads on the
blade as well as the blade structure were simulated. It furthermore describes the simulation model. Sec. 3 presents the results of the load and structural integrity calculations. Sec. 4 discusses the feasibility of the repair concept by suggesting pretensioning and stabilizing devices. Finally, a conclusion summarizes the findings of this research.

## 2 Methods

### 2.1 Up-tower repair concept

The proposed repair concept for a damaged pultruded CFRP spar cap comprises the following steps:

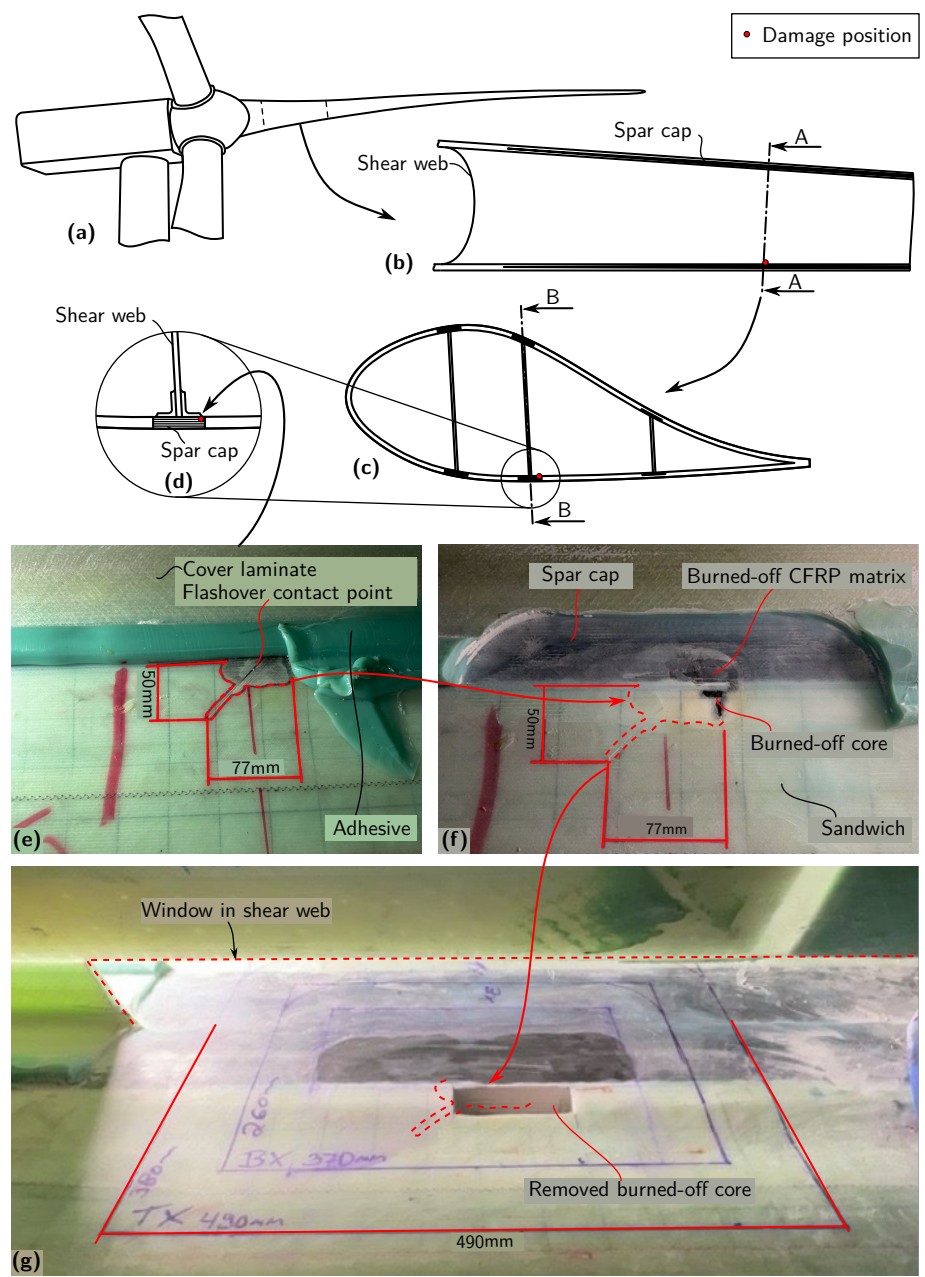

**Figure 2.** Up-tower repair concept: (a) wind turbine in rotor lock with damaged blade in 3 o'clock position, (b) longitudinal repair zone view through cut line B-B, (c) cross-sectional repair zone view through cut line A-A, (d) zoom into damage position, (e) damage observed as consequence of internal lightning flashover, (f) damage after grinding off cover laminate and adhesive, revealing burned-off spar cap matrix and core material, (g) damage after grinding off CFRP and window cut-out in shear web; based on Rosemeier et al. (2024); Lessa (2023).

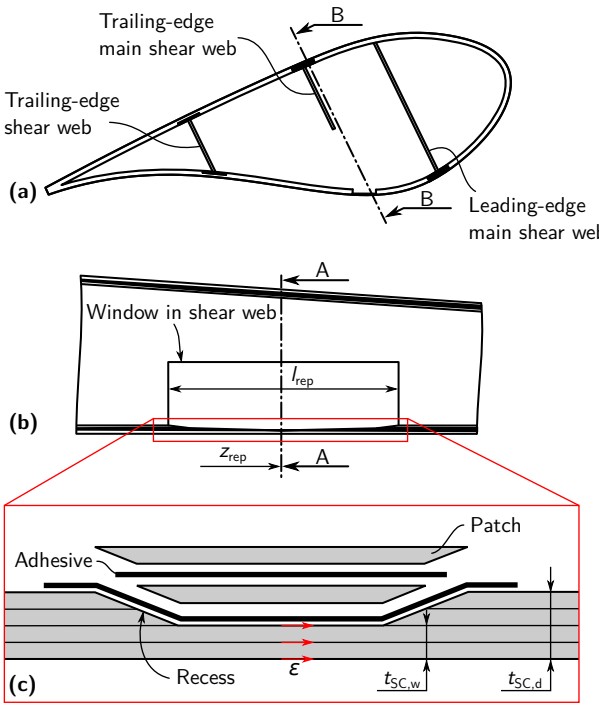

**Figure 3.** Weakened repair zone in (a) cross-sectional view through cut line A-A and (b) longitudinal view through cut line B-B; (c) exploded view of repair; based on Rosemeier et al. (2024) and Girolamo (2020).

Identify the depth of the spar cap damage, i.e., the number of damaged planks through the spar cap thickness. This can be done by a visual inspection, e.g., after a lightning strike on the rotor blade, by non-destructive testing (NDT) means such as ultrasonic testing (UT), or by destructive means, i.e., grinding off the CFRP planks until no damage is visible. The length of the repair zone $l_{\text{rep}}$, i.e., the recess and the web window, can then be determined (Fig. 3b).

Determine a rotor blade position such that the load on the weakened blade structure is minimized and the buckling resistance of the structure is maximized. A rotor blade position can be defined by a set of angles: the pitch angle, the azimuth angle, and the yaw angle (Fig. 4). This step can be conducted by means of aeroelastic simulations resulting in extreme load envelopes.

Determine whether the mechanical load and the mechanical resistance satisfy the condition that the rotor blade can remain connected to the wind turbine during the repair. Mechanical load refers to stress or strain, and mechanical resistance refers

to permissible strength, strain, and buckling resistance. This step can be carried out by means of finite element (FE) analysis using a shell model, for example.

When the load satisfies the permissible limits of the weakened blade, i.e., after any grinding/opening the repair area, the rotor blade can remain connected to the wind turbine, and an up-tower repair can be carried out. If the permissible limits are not satisfied, temporary support structures can be installed to reduce the loads or increase the buckling resistance of the repair

zone.

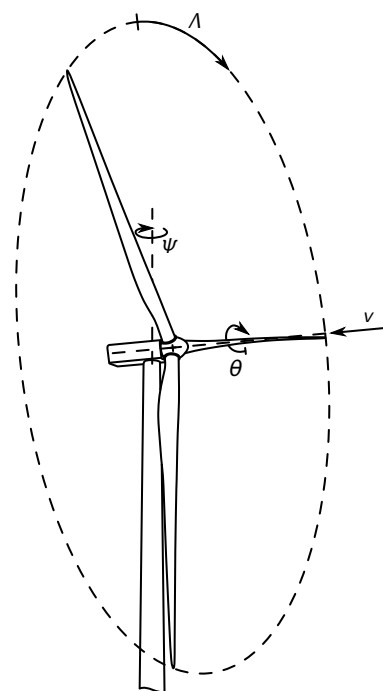

**Figure 4.** Repair position of rotor: yaw angle $\Psi$, azimuth angle $\Lambda$, and pitch angle $\theta$; mean wind speed direction $v$ shown normal to rotor plane; based on Rosemeier et al. (2024).

## 2.2 Aeroelastic load simulation of use case turbine

The MWT167H/7.0, also known as "SeaAngel", is a variable-speed, pitch-controlled wind turbine with a rated power of 7.0 MW. The turbine was designed for offshore applications under IEC IB conditions according to IEC 61400-3 (IEC, 2009). The average wind speed for the reference site is $v_{\mathrm{ave}} = 10.0\,\mathrm{ms}^{-1}$ and the reference turbulence intensity $I_{\mathrm{ref}} = 0.14$.

The blade geometry of the M-EU167 prototype was sliced into 52 cross-sections to obtain the airfoil coordinates. The airfoil polars were determined by Rfoil, an extension of Xfoil (Drela, 1989) which includes rotational effects (Bosschers, 1996). The airfoil polar calculations assumed a clean surface and a Reynolds number of $Re = 5 \cdot 10^6$.

The use case turbine was modeled in the wind turbine simulation software openFAST v2.4.0 (NREL, 2020). The aero-dynamic loads were calculated by AeroDyn (Moriarty and Hansen, 2005) taking the aeroelastic coupling with the bodies representing the tower and the blades into account. The aerodynamic properties included were determined as explained in Sec. 2.3. The turbulent wind fields were generated using TurbSim (Jonkman, 2009). Since the effect of coupled wave loads on the blades is small, ocean conditions have been neglected for this study. Hence, the tower was modeled with an onshore foundation. A generic tower model was derived based on similar offshore tower designs. The tower has a height of 109.8 m, yielding a hub height of 112 m. The tower is modeled as a flexible modal body using ElastoDyn and assuming the classic beam theory according to Euler-Bernoulli (Gere, 2004).

BeamDyn (Wang et al., 2014, 2016; Wang and Yu, 2017) was used to model the rotor blade structure. Here, an FE approach was used to take into account full structural coupling, i.e., the complete 6x6 stiffness and mass matrices determined using BECAS, as well as geometric non-linearity, whereby the computational costs were higher than those of the modal approach.

In the load simulation, the design load case (DLC) 8.1 according to IEC 61400-1 Ed. 4 (IEC, 2019a) was considered. The load case assumes an operating yaw system, the rotor in a locked state in the 3 o'clock position, and all blades pitched to feather. The generator is not connected to the grid. To find the load situation most damaging to the weakened blade, several average wind speeds from $3\,\mathrm{m\,s^{-1}}$ to $10\,\mathrm{m\,s^{-1}}$ were simulated. In addition, mean inflow inclinations of $0°$ and $8°$ with respect to the horizontal plane were considered. It was assumed that the design turbulence intensity covers the effective turbulence on site sufficiently well so that additional wake turbulence was neglected. For each average wind speed, six time series with a simulation time of $600\,\mathrm{s}$ were generated.

Since the turbine is under maintenance for the whole duration of the blade repairs, generator torque and pitch controllers were not included in the simulation.

The extreme loads were obtained as mean values of the extrema of the time series within each mean wind speed as required by IEC 61400-1 Ed. 4 (IEC, 2019a). Extreme load envelopes were determined as described by DNV A/S (2024) for the flapwise and lead-lag load directions.

## 2.3 Finite element analyses

A 3D FE shell model was generated in Ansys Parametric Design Language (APDL) with the help of the FEPROC toolbox (Rosemeier, 2018). The blade laminate was modeled using $27\,528$ elements, i.e., quadratic shell elements of type SHELL281, and quadratic solid elements of type SOLID186 for the trailing-edge adhesive layer, cf. Fig. 5b. The average element edge length was $\approx 280\,\mathrm{mm}$.

The FE shell model was subjected to mechanical extreme load situations stemming from the aeroelastic load simulations (Fig. 5a). The bending-moment distributions from the aeroelastic simulations were converted into shear forces $F_x$ and $F_y$ at 50 positions along the entire blade span. Rigid-body elements of type RBE3 transferred the forces from the master node(s), positioned along the blade reference axis, to the slave nodes positioned at the load-carrying structures, i.e., the CFRP spar caps, cf. Fig. 5b. Blade root nodes were constrained in all degrees of freedom. The repair zone was modeled by eliminating the web along the repair zone $l_{\mathrm{rep}}$ as well as reducing the spar cap thickness to $t_{\mathrm{SC,w}}$ in the middle, considering chamfer angles as a function of $l_{\mathrm{rep}}$, cf. Fig. 3.

A linear static analysis and a linear bifurcation analysis were conducted. An APDL routine extracted the buckling resistance and the longitudinal strains along the spar cap.

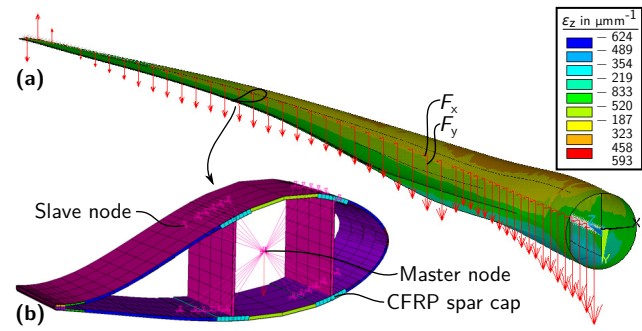

**Figure 5.** Finite-element model: (a) shear forces $F_x$, $F_y$ along blade span, and (b) rigid-body element (RBE3) transferring forces from master to slave nodes connected to the load-carrying structure, i.e., CFRP spar caps; color scale shows longitudinal strain $\varepsilon_z$; shell elements are shown as extruded.

## 3 Results

### 3.1 Repair situation

Bending moments due to gravity loads dominate strains in the repair zone under maintenance DLC. Consequently, rotor azimuth positions of $\Lambda = 90°$ and $270°$ were investigated, where $0°$ denotes the blade pointing upwards (Fig. 4). The $0°$ and $180°$ azimuth positions were not relevant as in this case, gravity mostly acts in longitudinal direction of the blade, while the aerodynamic loads contribute little to the loading. Hence the 6 o'clock position was not investigated further. While a yaw misalignment of $\Psi = 8°$ in both directions relative to the mean wind direction $v$ was considered, the highest loads occured at a yaw angle of $\Psi = 0°$. Vertical upflow of the wind field was taken into account for upflow angles of attack of $\alpha = 0°$ and $8°$ with respect to the horizontal plane, where an upflow of $0°$ led to higher loads. In all load cases, the blade was feathered at $\theta = 90°$ pitch.

Notably, as the wind speed increased, the mechanical loads on the rotor blade decreased. The aerodynamic design of the rotor blades meant that the resulting aerodynamic forces on the blade counteracted the gravity loads at rotor azimuth angles of both $90°$ and $270°$. With increasing wind speed, the aerodynamic forces increased, counteracting the gravity loads further. Hence, the most critical load case would occur at low wind speeds, where the mean aerodynamic uplift of the blade is smallest. Fluctuations in these loads as a result of turbulence were of minor importance in this context.

### 3.2 Structural response

This section describes the impact of the worst maintenance load envelope on the longitudinal strain and the buckling resistance of the weakenend blade structure. To this end, the length of the repair zone $l_{rep}$ was increased step-wise from $0\,\mathrm{m}$ to $5\,\mathrm{m}$ while the central damage position was chosen to be at a relative blade length of $z_{rep}/l_B$=17%, cf. Fig. 2 and Fig. 3. Within the length of $l_{rep}$, the original spar cap thickness of $t_{SC,d} = 41\,\mathrm{mm}$ was reduced to $t_{SC,w} = 26\,\mathrm{mm}$ while the shear web was removed

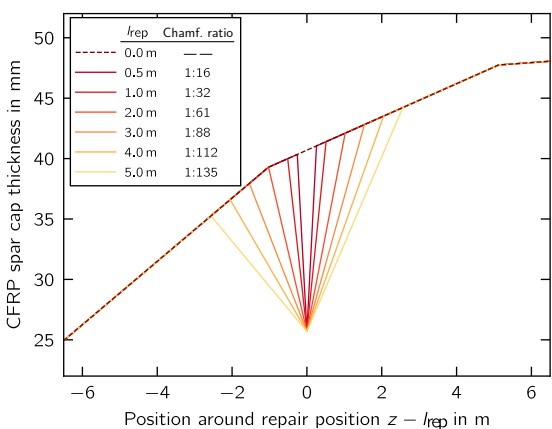

**Figure 6.** Spar cap thickness around repair zone of length $l_{rep}$ with corresponding chamfer ratios.

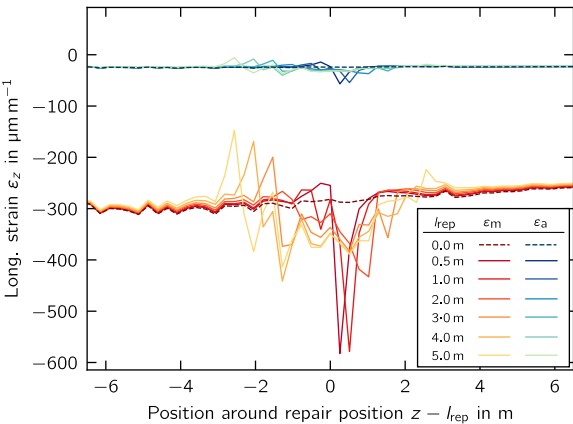

**Figure 7.** Longitudinal strain along weakened spar cap as function of repair zone length $l_{rep}$.

completely. The step-wise increase of the repair zone length decreased the chamfer ratio from 1:16 ($l_{rep} = 0.5\,\mathrm{m}$) to 1:135 ($l_{rep} = 5.0\,\mathrm{m}$), cf. Fig. 6.

Considering the longitudinal strain along the spar cap of the intact structure ($l_{rep} = 0\,\mathrm{m}$ in Fig. 7), the strain amplitude is relatively low, i.e., $\varepsilon_a \approx 25\,\mathrm{\mu m\,m^{-1}}$, when compared to the compressive mean strain of $\varepsilon_m \approx -300\,\mathrm{\mu m\,m^{-1}}$. When the structure was weakened, the strain amplitude increased toward $\approx 50\,\mathrm{\mu m\,m^{-1}}$ and the mean strain toward $\approx -600\,\mathrm{\mu m\,m^{-1}}$. The longer the repair zone $l_{rep}$, the smaller the strain increase, i.e., for $l_{rep} = 5\,\mathrm{m}$ the average mean strain was $\approx -350\,\mathrm{\mu m\,m^{-1}}$. Moreover, the strain rootward from the repair position fluctuated more than the strain tipward from the repair position.

Considering the buckling resistance as a function of the repair length (Fig. 8), the resistance reduces significantly from $P/P_{cr} = 20.7$ to $P/P_{cr} = 4.0$ when the web window is cut out for $l_{rep} = 2\,\mathrm{m}$ and the spar cap is weakened. For $l_{rep} > 2\,\mathrm{m}$

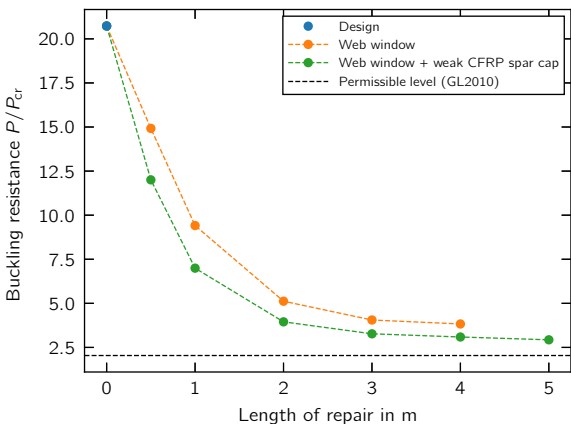

**Figure 8.** Buckling resistance.

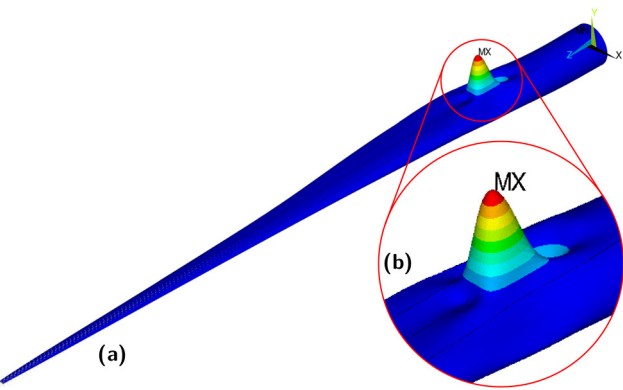

**Figure 9.** Buckling mode for 5 m long repair with the largest deformation indicated by MX.

the resistance reduces linearly with a relatively small change and approaches the permissible level. The largest deformation
magnitude of the buckling mode (Fig. 9) develops at $z_{rep}$ (Fig. 3b).

To investigate how the weakened spar cap affects the buckling resistance reduction, we calculated the buckling resistance for
the window cut-out only without including the weakening of the spar cap, cf. orange dots in Fig. 8. The removed trailing-edge
main web (window) and consequently wider panel between the leading-edge main web and the trailing-edge web (Fig. 3a) is
the major factor reducing the buckling resistance. At shorter repair lengths of less than 1 m, the weakened spar cap made a
135 larger contribution to the reduction, i.e., between 18% and 33%, than at longer lengths, where the figure was <7%.

## 4 Discussion

### 4.1 Structural integrity

The structural response shows that even quite small repair window lengths in the shear web, i.e., 1 m in length, along with the reduction in spar cap thickness to 37% can significantly reduce the buckling resistance of a blade subjected to the maintenance load case, i.e., an up-tower repair situation in 3 the o'clock azimuth position.

Moreover, the strain amplitude that may arise from wind turbulence was found to have a very low value for the blade structure of $\varepsilon_a \approx 25\,\mu\text{m}\,\text{m}^{-1}$ at wind speeds below $10\,\text{m}\,\text{s}^{-1}$. Although this value is very low, the matrix material and bonded joints must cure while these movements occur, and the permissible strain amplitude must be determined experimentally. However, the mean strain level, which is mainly a consequence of the gravity load in the 3 o'clock position, can increase by a factor of 2 depending on the window length. Our simulations showed that, for the use case turbine, lower wind speeds lead to higher compressive loads in the spar cap, as at higher wind speeds aerodynamic lift loads would partly outweigh the gravity loads. Therefore, it appears to be important to add a pretensioning structure which can be temporarily mounted and demounted up-tower, cf. Sec. 4.2. For the case that repairs may take longer than expected and also unexpected stronger winds may arise, the weakened blade could be temporarily rotated into the 6 o'clock position in order to eliminate at least the gravity loading in the repair position.

As the spar cap increases in thickness from rootward start to tipward end around the repair position, the recess introduces a dimple in the spar cap at the rootward start of the recess (Fig. 6). The dimple introduces a stiffness jump which explains why we observed larger strain fluctuations for longer repair lengths (Fig. 7), i.e., strain increase and strain decrease compared to the intact situation.

Girolamo (2020) suggested filling the recess of a spar cap made from CFRP planks using patches made from chamfered CFRP planks, cf. Fig. 3c. Other fabrics could also possibly be used. However, we would like to emphasize that it is also necessary to restore electrical properties adversely affected by damage, provided that the load-bearing structures exhibit such properties. For example, CFRP pultrusions are electrically conductive, which is why their use in rotor blades requires either electrical separation, i.e., through insulation and physical separation, from the lightning protection system of the wind turbine, or the provision of defined equipotential bonding connections with components of the lightning protection system, cf. IEC (2019b), such as lightning conductor cables and/or metal grid foils arranged on the surface of the rotor blade, cf. Thwaites et al. (2022). CFRP spar caps can also be used as a main down conductor along portions of the blade (Nauheimer and Ponnada, 2023). If equipotential bonding connections are provided or the spar cap is used as a main down conductor, any adverse effects on the the electrical conductivity of the load-bearing structure and/or the associated equipotential bonding connections must be remedied during the repair.

DNV A/S (2024) suggests a chamfer ratio of at least 1:150 for uni-axial CFRP materials below an areal weight of $1000\,\text{g}\,\text{m}^{-2}$. This results in a repair length of 4.5 m for the modeled spar cap thickness reduction of $t_{\text{SC,d}} - t_{\text{SC,w}} = 15\,\text{mm}$. As the aerial weight of CFRP planks is typically larger, DNV A/S (2024) suggests to determining the repair length by means of structural

tests of the repair laminate or assumptions using

$$l_{\text{rep}} = 2\frac{\sigma_{11,\text{t,rep}}}{\tau_{\text{int}}}\left(t_{\text{SC,d}} - t_{\text{SC,w}}\right)\gamma_{\text{proc}}\gamma_{\text{env}}, \tag{1}$$

where $\sigma_{11,\text{t,rep}}$ denotes the tensile strength of the repair laminate, $\tau_{\text{int}}$ the shear strength of the interface between repair material and recess, $\gamma_{\text{proc}} = 1.15\ldots1.3$ denotes the reduction factor for the repair process, and $\gamma_{\text{env}} = 1.0\ldots1.3$ the reduction factor for environmental influences. Assuming $\sigma_{11,\text{t,rep}} = 0.73\cdot2800\,\text{MPa}$ (Bussiba et al., 2022) with a restoration factor determined from a uni-directional glass FRP of 0.73 (Ghafafian et al., 2021) and $\tau_{\text{int}} = 12\,\text{MPa}$, a minimum window length of $l_{\text{rep}} = 5.9\,\text{m}$ to 8.6 m can be estimated. These estimates show that the ultimate strength of the repair depends significantly on the repair length, the interface to the recess, as well as the process and the environmental conditions. We therefore suggest that experimental investigations be conducted to characterize the load-carrying capability of CFRP plank repairs under up-tower repair conditions in order to provide guidance beyond the latest recommendations in standards and guidelines.

## 4.2 Support structure concepts

This section introduces support structure concepts that aim to solve problem 1 (P1), i.e., reduce the mean strain level along the spar cap during the repair, and problem 2 (P2), i.e., increase the buckling resistance during the repair.

To reduce the mean strain and address P1, a span-wise pretensioning structure as shown in Fig. 10 can be applied from the outside of the blade. The support structure comprises two clamping frames arranged exterior to the rotor blade and conceptualized to exert a clamping force on the rotor blade. The clamping frames are connected by a longitudinal support rod pretensioned to reduce the mechanical load, particularly the strain, on the load-bearing structure, i.e., the spar cap. Thus, the load-bearing structure can be relieved by clamping the rotor blade on both sides of the damaged area. However, installation concepts to lift the frames, e.g., into the 6 o'clock postion by winches, need to be developed. The clamping frames are spaced apart in the longitudinal direction of the rotor blade. Their concept is similar to that of load frames familiar from rotor blade testing, for example, where they are used to apply a load to the rotor blade, cf. Klaenfoth et al. (2023). Shear webs may require reinforcement to withstand the clamping force from the load frame. Each of the clamping frames includes a clamping jaw adapted to the outer contour of the rotor blade, which is arranged between two clamping beams. The clamping beams of each clamping frame are connected by two clamping elements to exert a force transferable to the rotor blade by the clamping jaw.

The clamping frames are connected to the longitudinal support rod by ball or universal joints. The longitudinal support rod includes pretensioning means (indicated as a thinner section of the rod in Fig. 10) for exerting a pretensioning force in the longitudinal direction on the rotor blade in such a way that the mechanical load, particularly the strain, on the load-bearing structure is reduced. The pretensioning means are adjustable to set the respective pretensioning force in the longitudinal direction, and are also conceptualized to adjust the length of the respective rod. The pretensioning means can include, for example, tension screws and/or turnbuckles and/or telescopic rods.

The support structure shown in Fig. 11 addresses P2 and consists of a support rod arranged inside the rotor blade with a transverse support rod to increase the mechanical resistance, specifically buckling resistance, of the load-bearing structure. For this purpose, the transverse support rod is connected with a clamping foot to the panel which requires the buckling support.

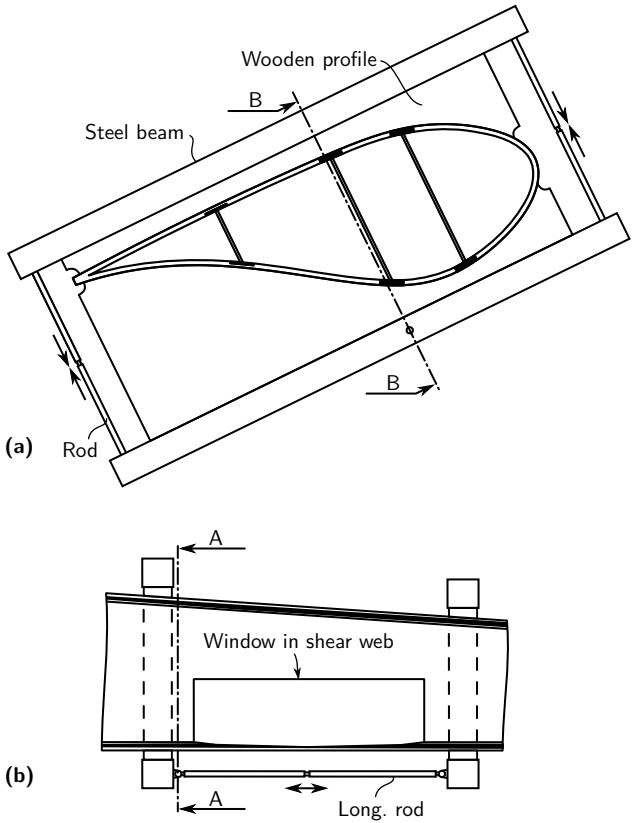

**Figure 10.** Span-wise pretensioning structure that solves P1: (a) cross-sectional view through cut line A-A and (b) longitudinal view through cut line B-B; based on Rosemeier et al. (2024).

Using additional clamping feet, the transverse support rod is connected to a panel opposite and to the web arranged above the damaged pultruded stack. The transverse support rod includes a pretensioning means to exert a transverse pretensioning force on the rotor blade in such a way that the buckling resistance is increased by supporting the buckling field of the panel, and thus

also the damaged pultruded stack of the adjacent load-bearing structure. The support structure may prevent buckling modes that develop toward the interior of the blade but not those that develop outward; it may even invert the buckling mode from inward to outward deformation. This drawback can be overcome by adding a buckling support structure on the outer surface, e.g., a stringer, as illustrated in Fig. 12.

Fig. 12 shows a support assembly which addresses P2 and is arranged interior or exterior to the rotor blade. The support

assembly includes one or more support profiles, also called stringers, arranged to increase the mechanical resistance, particularly buckling resistance, of the load-bearing structure and/or the non-load-bearing structure. The support profile consist of an FRP material, such as glass or carbon fiber, and/or metal. Shown are support profiles with an Z-shaped cross-section. Other cross-sections are possible, such as T-shaped, C-shaped, or L-shaped cross-sections. The support assemblies are connected to

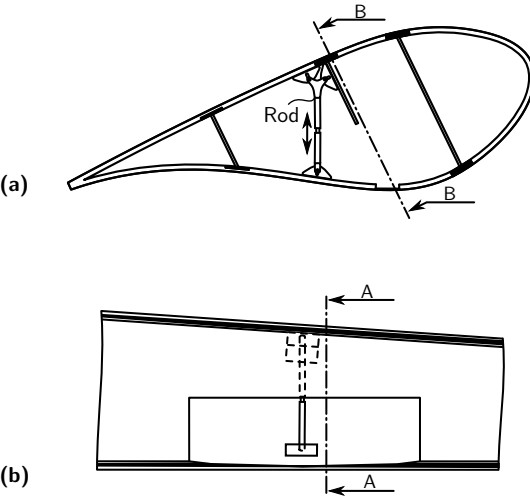

**Figure 11.** Buckling support structure - push rods that solves P2: (a) cross-sectional view through cut line A-A and (b) longitudinal view through cut line B-B; based on Rosemeier et al. (2024).

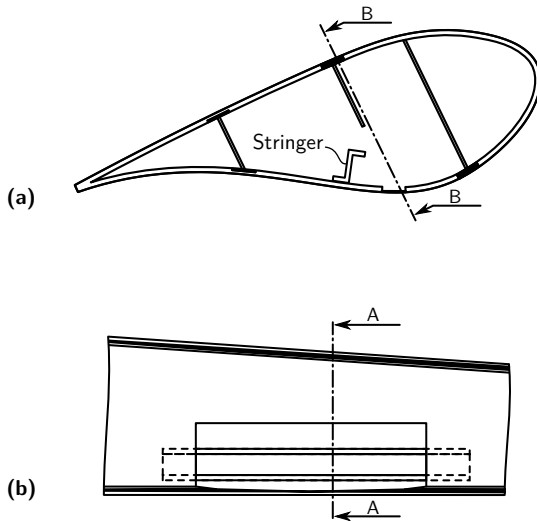

**Figure 12.** Buckling support structure - stringer that solves P2: (a) cross-sectional view through cut line A-A and (b) longitudinal view through cut line B-B; based on Rosemeier et al. (2024).

the interior of the rotor blade by a detachable adhesive connection, for example, using a thermoplastic material, or a magnetic
arrangement is provided for a simple, reversible, and repositionable connection.

Further suggestions for support structures that address the two problems P1 and P2 can be found in Rosemeier et al. (2024).

This research shows that the critical step during an up-tower repair of carbon spar cap planks may technically be feasible. From an economic point of view, we expect that the effort spent on any detailed development of structural support concepts

and a reliable method of mounting and demounting them will outweigh the effort that would be spent on a down-tower repair or even blade exchange including crane and transport.

## 5 Conclusions

This research identified the worst-case load effects and turbine conditions associated with up-tower repairs through comprehensive aeroelastic simulations. Our findings indicate that substantial repairs pose significant risks to structural stability. However, it was determined that strain amplitudes resulting from turbulence do not present critical concerns during these repair processes. To facilitate safe up-tower repairs, it is recommended that temporary pretensioning and buckling support structures be installed. These measures can effectively enhance the structural integrity of the blade, allowing for the repair of extensive damage without compromising safety. Furthermore, we proposed several innovative support structure concepts that can be installed from both inside and outside the blade, and are aimed at temporarily augmenting pretensioning and buckling resistance during repair activities. Overall, the insights gained from this research underscore the feasibility of conducting large-scale repairs in an up-tower setting while maintaining structural safety and integrity.

Future research should investigate the feasibility of the proposed up-tower repair concepts through experimental studies, such as those conducted in a blade test laboratory. This empirical approach will provide valuable insights into the practical application and effectiveness of the repair concepts identified. Of particular interest is the experimental study of the curing of matrix material and bonded joints while they are subjected to cyclic movements, in order to determine the permissible strain amplitude. Additionally, it is essential to explore options for equipotential bonding between CFRP repair patches and the CFRP planks in the recesses. One option could be the use of electrically conductive adhesives. Evaluating their structural fatigue strength will be crucial in ensuring the long-term reliability of the repairs. Furthermore, an investigation into the effects of chamfer ratios applied to CFRP plank repair patches is necessary. This research should focus on optimizing chamfer designs to minimize the length of the repair zone, thereby enhancing the efficiency and effectiveness of the repair process.

*Data availability.* The data presented in the figures is available at https://doi.org/10.5281/zenodo.15105994 (Rosemeier et al., 2025).

*Author contributions.* MR and MS modeled the use case turbine and its controller. CB conducted the aeroelastic load simulations. MR analyzed the modified blade structure. EP initiated this research and wrote the paper together with MR and MS. TK analyzed the measurement data. JK and PB evaluated the technical feasibility of the support structure concepts.

*Competing interests.* The authors declare that they have no conflict of interest.

*Acknowledgements.* We acknowledge the support provided by the German Federal Ministry for Economic Affairs and Climate Action (BMWK) within the CareUp project (03EE3109A). Moreover, we would like to thank Mitsubishi Heavy Industries, Ltd. and EUROS Entwicklungsgesellschaft für Windkraftanlagen mbH for providing the geometry and the laminate plan for the wind turbine blade model for this research. Finally, the authors would like to thank David Melcher, Bernd Haller, Christian Meyer, Tobias Holst, Sören Eden, Niels Ludwig, and Matthias Lindermann for their assistance in developing the support structure concepts.

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
