# Peer review of "Feasibility of up-tower repair concepts for pultruded carbon spar cap planks in wind turbine blades"

_Wind Energy Science, 2025_

## Author Response (AR1)

**AUTHORS' RESPONSE**

M. Rosemeier, M. Saathoff, J. Kolsch, P. Buriticá, E. Petersen,
and C. Beer

November 11, 2025

Dear Editor,

The manuscript wes-2025-56 entitled "Feasibility of up-tower repair concepts for pultruded carbon spar cap planks in wind turbine blades" submitted to Wind Energy Science, has been revised. This document summarizes the actions we have taken with respect to the reviewers' comments received until 2025-09-18.

We appreciate the reviewers' comments very much and are grateful for the helpful suggestions.
This document lists our actions and then repeats the reviewers' comments referring to the actions taken.
If not explicitly stated, the references to lines are valid with the attached document showing the differences between revison 1 submitted on 2025-05-19 and revision 2 (diff_main_rev02_vs_rev01.pdf).

With kind regards,

Malo Rosemeier

**1 ACTIONS**

A.01 (l.19, l.185f, l.190, l.200) Replaced "design" by "concept".

A.02 (l.20) Added reference to F2.

A.03 (F2) Increased pictures in F2 and added more descriptions, as well as scales.

A.04 (F3, caption) Corrected cut line references.

A.05 (l.49) Removed non-applicable wording.

A.06 (l.50f) Corrected wording.

A.07 (l.53f) Highlighted that permissible limits should include the weakened blade condition.

A.08 (l.93ff, new F5) Added more details and figure to the finite element modeling.

A.09 (l.71) Replaced latin reference with a modern english reference.

A.10 (l.112) Added note.

A.11 (Former F5, now F6.) Extended caption and improved legend readability.

A.12 (l.143f, l.236f) Added notes on generating permissible strain amplitudes during curing.

A.13 (l.168f) Corrected wording and unit.

A.14 (l.191f) Added note on clamping thorough load frames.

A.15 (l.208f) Added note on additonal buckling support.

A.16 (l.215) Corrected wording.

**2 COMMENTS**

**2.1 REVIEWER 1**

The topic of this paper is very interesting at least for O&M engineers and managers of wind farms on- and off-shore. The manuscript is well written and its content presented in a clear and unambiguous way, more or less (see also specific comments below).

RC1.01 The authors do not clearly state where the novelties of this work lie. The manuscript follows closely a patent application by the authors, published in 2024, and using even the same figures and arguments no scientific proof of concepts is provided. Section 4.2 "Support structure concepts" contains a sole presentation of brain- storming ideas without detailed calculations, implementation or experimental verification of their applicability. The attractive call for up-tower repairs mostly relies

on cost reduction compared to down-tower processes. Again, there is no proof, evidence or clue on whether proposed ideas, even if pragmatic, can lead to any cost reduction for O&M budgets. In light of the above, if the manuscript is published, it is better to drastically reduce or even remove Section 4.2 by quoting the relevant patent application. Also the title has to be modified to correctly reflect the content; some ideas are cast but it is uncertain whether they are technically feasible or even advantageous cost wise.

→ *We understand your main concern is the clarity of novelty. Our manuscript's intent is to contribute a structural feasibility assessment rather than to claim new field procedures or cost benefits. Specifically, we investigate feasibility via (i) simulations and (ii) practical considerations: an FE model of an 81.6 m commercial blade for a 7 MW offshore turbine under a maintenance load case, with loads from aeroelastic simulations, evaluating the weakened structure (step-wise increased repair zone) for buckling, cyclic strains, and permissible wind speeds for conducting the repair. We confirm that the support-structure content reflects brainstorming-concept proposals to enable up-tower repairs, not mature designs. These proposals should be developed further and validated in more detail. We also agree that cost and asset availability drive the decision, and we do not present proof of O&M cost reduction; as you note, no such evidence is provided or claimed.*

RC1.02 It is mentioned in the abstract: "...Various innovative support designs are proposed for installation..." but only a conceptual abstract sketch is shown for each idea. "Design" is more than that. Modify accordingly.

→ *See A.01.*

RC1.03 Section 2.1, Line 41: " ...The proposed repair concept ..." The authors must clearly state that this type of up-tower repair in CFRP spar caps by cutting also windows in shear webs is already known and performed in the field by a few repair companies. It is not an innovative solution presented by the authors. There are no related publications because these are technical solutions and processes belonging to the commercial and not the academic universe of wind energy technology

→ *Our focus is not to claim novelty for individual repair steps - which are already performed by repair companies (as cited in F2) - but to combine established practices into a coherent up-tower sequence and assess its structural feasibility (limits) for pultruded CFRP spar caps under maintenance loads.*

RC1.04 Line 46: for what purpose " ...Determine a rotor blade position such that the load on the weakened ..." since to repair the blade it should remain horizontal, i.e. at 3 or 9 o'clock? It could be perhaps proposed to determine by aeroelastic calculations this optimal position of the damaged blade for harsh wind conditions but as for repair works, must be horizontal.

→ *We define a rotor blade position not only by azimuth but also by pitch angle, and yaw angle. In some cases a repair position of 6 o'*

*clock can also be benefitial. This position can be found by aeroelastic simulations, as explained.*

RC1.05 Section 2.3, Line 92-93: "...The resulting shear force distribution was transferred from the load simulation to the FE shell model using rigid-body elements of type RBE3..." Describe this in detail; how exactly "shear force distribution" is transferred? In the entire blade model or in segments? How many RBE's are introduced? By "shear force distribution" the discrete force and moment vectors developed at the nodes of the aeroelastic beam model are meant? As vaguely described it leaves suspicions on static equivalency.

→ *See A.08.*

RC1.06 Fig.5 is incomprehensible and text in Lines 115-118 is not helpful.

→ *We have tried to identify which parts were incomprehensible in F5 and which text was unhelpful in lines 118–155. However, we did not find any missing information. We agree that the figure requires some time to digest, but we believe it is acceptable for a scientific journal article. See A.11.*

RC1.07 Section 4.2: As already argued, the information contained here can be recalled in the related patent application and since no proof of concept, whatsoever, analytical, computational, experimental is presented could be removed.

→ *Our intent is to keep it as part of the discussion because it documents the proposal space (and is only an excerpt of the related patent application), which helps contextualize the feasibility analysis.*

RC1.08 Conclusions: The two first lines are contradicting with the content of the manuscript; the determination of optimal position of the damaged blade depending on wind conditions is irrelevant since repair is only possible when the blade is in horizontal position

→ *Firs two lines: "This research identified the worst-case load effects and turbine conditions associated with up-tower repairs through comprehensive aeroelastic simulations. Our findings indicate that substantial repairs pose significant risks to structural stability."*
*Regardless of whether the repair position is horizontal, we identified the worst-case loads at such a repair position, which can significantly reduce the buckling resistance. As this is one of our main contributions/findings, these two lines appear to agree with the content of our manuscript.*

RC1.09 Line 21: reference to Fig.3 in the text precedes that to Fig.2

→ *See A.02.*

RC1.10 Caption of Fig.3: "...cut line B-B"; replace by A-A, "...cut line A-A"; replace by B-B

→ *See A.04.*

RC1.11 Line 70: Euler-Bernoulli classic beam theory is more than enough! Reference to a 3 centuries old latin source (Euler L. 1744) is an exaggeration.

→ *See A.09.*

RC1.12 Line 161: replace "aerial" by "areal", replace "kgm-2" by "gm-2"

→ *See A.13.*

**2.2 REVIEWER 2**

Thank you for this interesting submission that already has decent quality.

The submitted manuscript addresses an interesting question with regard to feasibility of up-tower repairs for wind turbine rotor blades. Most of my comments address minor revisions in the manuscript. However, one major problem of the underlying technical challenge has not been addressed.

RC2.01 (F2) Pictures 2.e, 2.f and 2.g are very small. It is very hard to identify, what you are trying to show. Especially the repair area in the pultrusions does no look like a professionally prepared repair chamfer (way too short to my feeling) is it possible to add a scale somewhere?

→ *In these example images, the CFRP damage does not appear to be as deep as could occur in the field; therefore, a short chamfer was used. We could not determine whether it was too short because we had no information on the measured CFRP damage depth. These images were selected for illustrative purposes only. See A.03.*

RC2.02 (l.49f) "can remain operational": I think, all of this section is rather about being safe than about being operational. Obviously, serviceability has been compromised by a damage which is why a repair needs to be carried out.

→ *See A.05.*

RC2.03 (l.50f)"a permissible strain or stress, i.e., strength, and mechanical resistance refers to a permissible buckling resistance ": From my perspective it is misleading to subdivide permissible strain and stress against mechanical load and resistance in connection with buckling. All of these can be expressed in terms of permissible stresses and strains. In my opinion it is a strengh-, strain- and buckling resistance.

→ *See A.06.*

RC2.04 (l.53f)"When the load satisfies the permissible conditions": This needs to explicitly consider the condition of the blade after grinding/opening up the repair area! Prior to verifying this condition no grinding operations should be carried out.

→ *See A.07.*

RC2.05 (l.107f)"The aerodynamic design of the rotor blades meant that the resulting aerodynamic forces on the blade counteracted the gravity loads at rotor azimuth angles of both 90° and 270°. With increasing wind speed, the aerodynamic forces increased, counteracting the

gravity loads further.": Seems to me as if some of the statements here are redundant.

→ *The first sentence should state the fact, and the second and third sentences should explain it; from our perspective, it seems okay. See A.10.*

RC2.06 (l.124f) "Considering the buckling resistance as a function of the repair length (Fig. 7), the resistance reduces significantly from P/Pcr = 20.7 to P/Pcr = 4.0 when the web window is cut out for lrep = 2 m and the spar cap is weakened. For lrep ¿ 2 m the resistance reduces linearly with a relatively small change and approaches the permissible level. The largest deformation magnitude of the buckling mode (Fig. 8) develops at zrep (Fig. 3b).":

→ *You can obtain the numbers from the graph in F7. I hope this helps.*

RC2.07 (l.183ff) " Their design is similar to that of load frames familiar from rotor blade testing, for example, where they are used to apply a load to the rotor blade, cf. Klaenfoth et al. (2023).": This is debateable since usually for example the shear web is reinforced in areas, where the load clamps are attached during blade testing. Therefore the test blade usually differs from the series blade. The introduced loads from the clamping rig as well need to be evaluated with regard to the specific rotor blade design.

→ *See A.14.*

RC2.08 (l.195f) "The support structure shown in Fig. 10 addresses P2 and consists of a support rod arranged inside the rotor blade with a transverse support rod to increase the mechanical resistance, specifically buckling resistance, of the load-bearing structure. ": The support structure may introduce an areal loading of the shells potentially initiating a post critical stability failure by inverting the buckling mode from deformation inward to deformation outward of the blade. Is there any means to prevent from this?

→ *See A.15.*

RC2.09 (l.204)"S-shaped": Z-shaped?

→ *See A.16.*

RC2.10 From my perspective one major problem is not addressed. The rotor blade structure (even if reinforced by above measures) will move and deform. Therefore bonding connections and laminations between planks or spar cap and spar web will move relative to each other. This will compromise the necessary bonding connections. Therefore such blades with critical damages to the main load carrying structural members are taken down and continuously supported to fix position and shape. These measures are not taken due to strength and or stability considerations up-tower.

→ *See A.12.*

**2.3 REVIEWER 3**

RC3.01 Die Erfahrung aus meiner Praxis zeigt, daß Reparaturentscheidungen aufgrund mehrdimensionaler Kriterien erfolgen, die nicht alleine technischer Natur sind. Die Verfügbarkeit von Ersatzteilen, Material und geeignetem Personal, sowie meteorologische Bedingungen spielen hier mit eine Schlüsselrolle. Gleichwohl zeigt das Verbundprojekt die Leistungsfähigkeit der Simulation und ingeniöse Kreativität durch versteifende Komforten als Exosketett oder Endoprthese die erforderliche Positionierung der Fügepartner herzustellen. Welch ein Aufwand am hängenden Blatt diese Hilfsvorrichtungen ohne Kran aus der Bühne oder dem Seil anzubringen.

$\rightarrow$ *We fully agree that repair decisions are multi-criteria and not purely technical. Your remarks on the considerable effort to install stiffening aids as an exoskeleton or endoprosthesis without a crane - from a platform or on ropes - highlight the operational challenges. Likewise, the choice between up-tower and down-tower repair is strongly context-dependent: offshore it hinges on the availability, mobilization time, and day rates of jack-up vessels, as well as suitable metocean windows; onshore it is shaped by the accessibility of remote sites (road and crane access, crane logistics, permits, seasonal constraints).*

**2.4 EDITOR COMMENT**

EC.01 Content of the manuscript submitted is in line with the scope of the journal. The methodology in terms of repair method and FEA should be elaborated further during a next review to provide more details and safeguard repeatibility to the readers.

$\rightarrow$ *See A.08.*